# Vitamin B_12_ Status and Optimal Range for Hemoglobin Formation in Elite Athletes

**DOI:** 10.3390/nu12041038

**Published:** 2020-04-09

**Authors:** Jarosław Krzywański, Tomasz Mikulski, Andrzej Pokrywka, Marcel Młyńczak, Hubert Krysztofiak, Barbara Frączek, Andrzej Ziemba

**Affiliations:** 1National Centre for Sports Medicine, Żwirki i Wigury 63A, 02-091 Warsaw, Poland; jarek.krzywanski@coms.pl (J.K.); hubert.krysztofiak@coms.pl (H.K.); 2Mossakowski Medical Research Centre, Polish Academy of Sciences, Adolfa Pawińskiego 5, 02-106 Warsaw, Poland; tomik@imdik.pan.pl; 3Department of Biochemistry and Pharmacogenomics, Faculty of Pharmacy, Medical University of Warsaw, Banacha 1, 02-097 Warsaw, Poland; a.pokrywka@coms.pl; 4Institute of Metrology and Biomedical Engineering, Faculty of Mechatronics, Warsaw University of Technology, św. Andrzeja Boboli 8, 02-525 Warsaw, Poland; mlynczak@mchtr.pw.edu.pl; 5Department of Sports Medicine and Human Nutrition, Institute of Biomedical Sciences, University of Physical Education in Krakow, al. Jana Pawła II 78, 31-571 Kraków, Poland; barbara.fraczek@awf.krakow.pl

**Keywords:** vitamin B_12_ status, athletes, hemoglobin, vitamin B_12_ threshold, injections

## Abstract

Background: Athletes and coaches believe in the ergogenic effect of vitamin B_12_ (which results from enhanced erythropoiesis) and they often insist on its unjustified supplementation. Therefore, our study aimed to assess the vitamin B_12_ status in Polish elite athletes and its influence on red blood cell parameters. Methods: In total, 1131 blood samples were collected during six years from 243 track and field athletes divided into strength and endurance groups, as well as according to the declared use of vitamin B_12_ injections. Results: An average vitamin B_12_ concentration in all subjects was 739 ± 13 pg/mL, with no cases of deficiency. A weak but significant relationship was found between vitamin B_12_ and hemoglobin concentrations. A significant increase in hemoglobin appeared from very low vitamin B_12_ concentration and up to approx. 400 pg/mL, while hemoglobin did not significantly change from 700 pg/mL and onwards. Vitamin B_12_ injections were used by 34% of athletes, significantly more often by endurance than by strength athletes. In athletes who declared no use of injections, a higher concentration of vitamin B_12_ was observed in the endurance group. Conclusion: The main finding of the present study is the determination of the range of vitamin B_12_ concentration which may favor better hemoglobin synthesis in athletes. They should regularly monitor vitamin B_12_ concentration and maintain the range of 400–700 pg/mL as it may improve red blood cell parameters. We might suggest application of a supplementation if necessary. Special attention is required in athletes with a vitamin B_12_ concentration below 400 pg/mL.

## 1. Introduction

Vitamin B_12_ (cobalamin) is a general name of several cobalt-containing corrinoid compounds, which are essential for the proper metabolism and function of all animal organs and systems. In the human body, only two forms are biologically active, adenosylcobalamin and methylocobalamin, functioning as coenzymes in radical-induced rearrangements and methylation processes, respectively. Adenosylcobalamin is a cofactor for mitochondrial methylmalonyl-CoA mutase and it is essential for the metabolism of fatty acids and ketogenic amino acids. Methylcobalamin is a cofactor for cytosolic methionine synthase that catalyzes the conversion of homocysteine to methionine. Methionine is then converted to S-adenosylmethionine, a universal methyl group donor for methylation reactions throughout the body, including the methylation of DNA, RNA and proteins [1,2,3].

Cobalamin is a unique vitamin produced exclusively by bacteria. It enters the animal food chain via herbivores, which accumulate cobalamin during intestinal fermentation of grass (performed by certain cobalamin-producing bacteria) [1].

The major physiological action of vitamin B_12_, relevant to sportsmen, include its involvement in red blood cells formation in bone marrow [4]. Another potentially beneficial actions of vitamin B_12_ are maintenance of proper immune function, improved transmission of neural signals and synthesis of neurotransmitters and creatine [5,6,7,8]. Athletes and coaches believe that enhanced red blood cell parameters are desirable for optimal performance, therefore hemoglobin concentration is their favorite biomarker. Thus, B_12_ is a commonly used supplement in many branches of sport and, in addition, many athletes and coaches strongly insist on the unjustified administration of vitamin B_12_, especially as injections [9,10,11,12,13,14,15,16].

A total serum or plasma vitamin B_12_ concentration is commonly used as the first line biomarker of deficiency, although it lacks sensitivity and specificity. Moreover, there is no consensus on the blood vitamin B_12_ lower cut-off point, which ranges from 100 to 350 pg/mL, and deficiency may occur even when the total serum vitamin B_12_ is within the normal range [17,18,19,20].

In a typical population of young adults, vitamin B_12_ deficiency is present in approximately 6%, and this figure largely depends on the region and race (is higher in Asia and Africa), but not on gender [21]. The deficiency incidents also increase with age, as well as in individuals with gastrointestinal disorders, vegans and chronic patients undergoing therapy with proton pump inhibitors, H2 antagonists or metformin [22,23,24]. Some of the aforementioned risk factors may also occur in athletes.

Despite vitamin’s B_12_ popularity in sport, the studies on its status and influence on performance in athletes are astonishingly limited and inconclusive [25,26,27]. There are virtually no studies on the relationship between vitamin B_12_ status and red blood cell parameters in athletes.

The aim of the study was to identify whether there is a relationship between serum vitamin B_12_ concentration and red blood cell parameters and, if so, to attempt to establish the reference range of vitamin B_12_ for athletes which provides the optimal hemoglobin formation.

## 2. Materials and methods

The study protocol was approved by the ethics committee of the Central Clinical Hospital of the Ministry of the Interior in Warsaw (permission number 134/2017).

The total number of 1131 fasting blood samples were collected from 243 track and field athletes, covering 904 samples from 189 strength athletes (age 23 ± 1 yr., body mass 79 ± 2 kg, height 181 ± 1 cm, BMI 24.5 ± 0.6 kg/m^2^) and 227 samples from 54 endurance athletes: (age 25 ± 1 yr., body mass 61 ± 1 kg, height 175 ± 1 cm, BMI 19.9 ± 0.2 kg/m^2^). The samples were obtained during the routine blood monitoring carried out in the National Centre for Sports Medicine over the period of 2009–2015.

Both the strength and endurance athletes were further subdivided into the following groups according to the declared use of vitamin B_12_ injections:

S0–strength athletes (throws, jumps and sprinters), who declared no use of vitamin B_12_ injections within the last three months (176 persons, 727 samples);

S1–strength athletes, who declared the use of vitamin B_12_ injections within the last three months (57 persons, 177 samples);

E0–endurance athletes (800 m and longer distance runners and race walkers), who declared no use of vitamin B_12_ injections within the last three months (40 persons, 143 samples);

E1–endurance athletes, who declared the use of vitamin B_12_ injections within the last three months (25 persons, 84 samples).

During the six-year period of the samples collection, 55 subjects (44 strength and 11 endurance) fell into both groups: using (S1/E1) or not using (S0/E0) vitamin B_12_ injections and in 53 subjects (42 strength and 11 endurance) the results of vitamin B_12_ concentration were available both before and after the injection.

Athletes were also asked if they believed that the use of vitamin B_12_ enhanced their performance. The obtained answers were as follows: 1–it definitely does not, 2–not really, 3–I do not know, 4–I suppose it does, 5–it definitely does. The answers 1–3 were defined as weak belief, whereas 4–5 as strong belief in the vitamin B_12_ influence on athletic performance.

Fasting blood samples were collected in the morning from the antecubital vein into the shaded Vacuette tubes with CAT serum clot activator (Greiner Bio-One, Monroe, NC, USA). All the blood parameters were determined by a certified laboratory company, Diagnostyka (Poland). Vitamin B_12_ concentration was determined in serum with an electrochemiluminescent method on Cobas E analyzer with a commercial kit by Roche. Linearity of the method is claimed to be within 50 and 2000 pg/mL, the lower detection limit was assumed at 150 pg/mL, the values above the 2000 pg/mL were not quantified but classified as >2000 pg/mL. The red blood cell parameters were determined with Sysmex XT 2100 hematologic analyzer, hemoglobin (Hb) by SLS photometry, hematocrit (Ht), mean corpuscle volume (MCV) and mean corpuscle hemoglobin (MCH) calculated from measurements by impedance with the hydrodynamic focusing method.

In the present paper, vitamin B_12_ concentrations are presented as pg/mL (as often used in clinical practice), whereas in some papers cited in discussion pmol/L are used (1 pg/mL = 0.7378 pmol/L) [1].

### Statistics

The analysis was performed using R software (version 3.5.0) and the fitting using Matlab 2019b (Mathworks Inc., Natick, MA, USA). The data are presented as mean ± standard error of the mean (SEM) and the level of significance was set at *p* < 0.05. The impact of B_12_ supplementation and the differences between groups E0/S0 and E1/S1 were analyzed using an unpaired Wilcoxon rank test (as all distributions were tested for normality using Shapiro-Wilk test, and they appear not to be normal). Moreover, a Kolmogorov-Smirnov test was used to test the differences in the distributions of strength and endurance groups. The relationships between B_12_ concentration and Hb, Ht, MCV and MCH parameters were evaluated using Pearson’s correlation test and linear regression models, where the intercept and slope were calculated.

In search for the reference range of vitamin B_12_ concentration for athletes, which provides the optimal red blood cell formation, two statistical approaches were applied. The first model is based on fit described using the equation:(1)$$Hb=Hb_{0}+\frac{\left(Hb_{M}-Hb_{0}\right)}{\left(1+\frac{K}{B_{12}}\right)}$$
where *Hb* and B_12_ stand for dependent and independent variables (y and x, respectively); and $Hb_{0}$ is the apparent hemoglobin when B_12_ = 0; $Hb_{M}$ is the maximal/saturated level of hemoglobin; K is the apparent half-response coefficient. The settings for the fitted curve were as follows: nonlinear least square method, bisquare scheme, Levenberg-Marquardt optimization algorithm and without establishing the limits for $Hb_{0}$, $Hb_{M}$, and K. In the second method, all samples were continuously divided (floating cut-off) according to different vitamin B_12_ concentration thresholds from the whole range of observed vitamin B_12_ concentration values. For every division point, the *p*-value for the difference in hemoglobin concentration was calculated using the unpaired Wilcoxon rank test. It produced the graph with a borderline *p* < 0.05, below which the differences are statistically significant and above which they are not.

In the group of 53 subjects (11 in Group E and 42 in Group S) for which the results of vitamin B_12_ concentration were available both before and after injection, the influence of the injection on the investigated blood parameters, Hb, Ht, MCV and MCH, were analyzed by paired Wilcoxon rank test.

## 3. Results

According to the laboratory’s normal ranges, no cases of vitamin B_12_ deficiency (<197 pg/mL) were identified; the average vitamin B_12_ concentration in all subjects was found to be 739 ± 13 pg/mL (703 ± 15 pg/mL in strength (range 205—>2000 pg/mL) and 881 ± 32 pg/mL in endurance athletes (range 242—>2000 pg/mL)), significantly higher in the endurance group (*p* < 0.001). The number and percentage of the strength and endurance athletes with vitamin B_12_ concentration below 300, 350 and 400 pg/mL and above 700 pg/mL are presented in Table 1. Significantly more samples with vitamin B_12_ concentrations below 300, 350 and 400 pg/mL were collected from the strength athletes, whereas the concentration above 700 pg/mL was more frequent in the endurance athletes. Additionally, the cumulative distributions between strength and endurance groups were compared by Kolmogorov-Smirnov test (not shown) and the *p*-value was <0.001, which means the two distributions are significantly different in terms of their shape.

Weak but statistically significant, positive relationships were found between vitamin B_12_ concentration and hemoglobin concentration (*p* < 0.001), hematocrit (*p* < 0.01) and MCH (*p* < 0.05); no correlation with MCV was observed (Figure 1).

The linear regression shown in Figure 1 was used to simplify the preliminary analysis, which aimed to establish the blood marker most responsive to changes in vitamin B_12_ concentration. It appeared to be hemoglobin, and this marker was subjected to additional analysis, considering a possible nonlinearity of this dependency.

The nonlinear plot in Figure 2 presents the calculation of mean ± SEM for the spans of 50 B_12_ units (for values up to 1000 pg/mL), and then for 200 B_12_ units wide sections (above 1000 pg/mL), and then the saturation curve was fitted (*Hb*_0_ = 0.4254, *Hb_M_* = 14.59, *K* = 5.076). Assuming an arbitrary threshold of 99% of the fitted saturated level, it corresponded to B_12_ = 488 pg/mL (Figure 2).

The Wilcoxon *p*-value for the significance of differences in hemoglobin concentration in samples, continuously divided according to different vitamin B_12_ concentration (floating cut-off), is shown in Figure 3. A significant difference in hemoglobin concentration was observed when the samples were divided by the borderline vitamin B_12_ concentration of 395 pg/mL. The significance increased up to vitamin B_12_ concentration of approximately 700 pg/mL and ceased to change further with the increase in vitamin B_12_ concentration. However, it should be noted that the interpretation of floating cut-off analysis should be considered with some caution. Indeed, such analysis is affected by clustering of the points on the surface, which changes from dataset to dataset and is irrelevant for the overall dependence of Hb on B_12_.

Vitamin B_12_ injections were used by a total of 34% of athletes (82 subjects), significantly more often by endurance athletes (46% of the endurance group, 25 subjects) than by strength athletes (30% of the strength group, 57 subjects), *p* < 0.05.

Analyzing the athletes’ belief in the ergogenic effect of vitamin B_12_ supplementation, significantly more endurance (54%) than strength athletes (38%) had a strong belief in the ergogenic effect of vitamin B_12_ (*p* < 0.05). Among the samples of athletes with a strong belief in the ergogenic effect of vitamin B_12_, the injections were used in 34% of samples, whereas in those with a weak belief this figure amounted to 13%.

A significantly higher vitamin B_12_ concentration was found in the athletes who used injections, both in strength and endurance groups (*p* < 0.001, Figure 4). In athletes who declared no use of injections, a higher concentration of vitamin B_12_ was observed in the endurance group (576.8 ± 11.4 pg/mL for S0 vs. 698.1 ± 27.3 pg/mL for E0, *p* < 0.001). There was no difference in vitamin B_12_ concentration between strength and endurance groups in athletes who used injections.

In 53 athletes (11 endurance and 42 strength), in whom plasma vitamin B_12_ concentrations were determined both without and after the use of injections, a significantly higher Hb and Hct were found only in the endurance group (*p* < 0.05; Figure 5).

## 4. Discussion

The main finding of the present study is the determination of the range of vitamin B_12_ concentration which may favor a better hemoglobin synthesis in athletes. The obtained results showed no vitamin B_12_ deficiency in athletes, at least defining the latter according to a frequently used lower limit of approx. 200 pg/mL. Such findings are consistent with majority of the studies conducted in athletes and highly active people [9,25,26,27,28,29]. Yet, a few contradicting reports describe some cases of deficiency in similar groups [14,30]. Our result is compliant with data concerning the vitamin B_12_ intake in athletes, which ranges from 2.5 µg/d to 11 µg/d, being at a lower level in vegetarians and vegans, whereas the recommended daily intake of cobalamin in adults is 2.4 μg/day [31,32,33,34,35,36]. In a study conducted in Polish endurance athletes, the intake of B_12_ < 2.4 μg/day was recorded in only 4% of the study participants, while the average consumption was twofold higher than the recommended dose [37]. Similar results were obtained by Fraczek in 80 Polish athletes, where 2.5% of subjects did not achieve the level of Estimated Average Requirement, whereas the excessive intake of cobalamin was observed in 96.5% [38].

The lack of cases of deficiency among the participants of the present study might result from the nutritional education and monitoring programs introduced in the elite athletes. The second issue is a common use of sport supplements, the majority of which contain vitamin B_12_.

In the present study, the significant differences in vitamin B_12_ concentration were found between strength and endurance groups. A higher average vitamin B_12_ concentration was recorded in endurance athletes, as well as more samples above 700 pg/mL were obtained from this group. A more frequent use of injections by endurance athletes vs. strength (46% vs. 30%, respectively) is potentially the main reason for such outcomes. The endurance athletes, who did not use injections, were also found to have higher vitamin B_12_ concentrations than in the strength group, which might result from a higher dose of vitamin B_12_ consumed in oral supplements. The possible explanation for such a finding is that both groups receive prescribed oral supplementation of multivitamins with the same vitamin B_12_ content, 23 µg/d. The endurance athletes, however, use higher amounts of sport drinks containing a set of vitamins, which may account for the additional 6 µg/d of B_12_, which is what, in our opinion, may be responsible for the observed difference between strength and endurance athletes.

In the present study, weak positive correlations between vitamin B_12_ concentration and Hb, MCH and Hct were found. Moreover, in endurance athletes, higher values of hemoglobin and hematocrit were observed after B_12_ injections.

The available data on relationship between B_12_ and hemoglobin were obtained from the sedentary and elderly subjects and, astonishingly, the influence of vitamin B_12_ supplementation on red blood cell (RBC) parameters is equivocal. Vitamin B_12_ supplementation, according to the guidelines, has been confirmed to restore RBC parameters in vitamin B_12_ deficiency anemia [39,40,41]. Yet, when assessing the influence of vitamin B_12_ on RBC parameters in the absence of anemia, the available data are inconclusive. Several studies have confirmed the relationship between vitamin B_12_ concentration and Hb, MCV, MCH and Hct [42,43,44,45]. However, the majority of studies [46,47,48,49], as well as the recent meta-analysis [50], failed to prove such a relationship.

The possible reasons are as follows: (1) the deficiency was not diagnosed properly on the basis of blood B_12_ status—there was no real deficiency in the tissues; (2) hemoglobin levels did not increase after the supplementation of non-deficient subjects—the effect can be observed only in the deficient subjects; (3) the decrease in hemoglobin did not occur because of the B_12_ deficiency alone—other coexisting genetic or environmental factors are necessary (e.g., inflammation) [50].

There are no data in young and very active people like the subjects of the present study. The obtained results rather support the thesis that higher vitamin B_12_ concentration enhances hemoglobin and this subject presents some interest for the further investigations. In athletes, especially of endurance sports, one of the adaptive changes to exercise is hemodilution [51,52,53]. Therefore, in those subjects an increase in hemoglobin induced by enhanced vitamin B_12_ status might not be visible in an ordinary blood hemoglobin measurement. Consequently, evaluation of the total hemoglobin mass should be considered in future studies [54,55].

The serum cobalamin is the first line and most common marker of vitamin B_12_ status, but its value is limited due to low sensitivity [17,19,56,57,58,59]. There is also no clear consensus concerning the exact value of the blood vitamin B_12_ concentration reference range, both in non-athletic population and in athletes [17,19]. In the available literature, the lower cut-off point for the blood vitamin B_12_ concentration ranges from 100 to 350 pg/mL [18,20]. Up to 45% of subjects might have a vitamin B_12_ deficiency, while having the total serum vitamin B_12_ within the normal range [5,60,61]. In some papers, a vitamin B_12_ concentration of 150–350 pmol/L is defined as the “grey zone”, and in such cases the introduction of additional markers, namely holotranscobalamin II (holoTC II) methylmalonic acid (MMA) and homocystein (Hcy), might solve the problem [17,62,63]. Unfortunately, no exact reference range for these markers has been determined so far [56,64,65,66,67,68].

In the present study, the concentration of 200 pg/mL was assumed as a lower limit of the normal concentration of vitamin B_12_ in blood serum. When such criteria was applied, no cases of deficiency were found. Nevertheless, 50 samples (4%) were below B_12_ = 300 pg/mL and 60 (6%) between 300 and 350 pg/mL, which mainly pertains to the strength athletes. Therefore, 10% were in the “grey zone” or below [17,20,62,63]. This raises the question of whether the grey zone should be extended to 400 pg/mL. Assuming such criteria, 18% of the samples were below that level, mostly strength athletes who did not use injections. Since the strength athletes are not reported to have vitamin B_12_ concentration different from non-athletic controls [69], a similar percentage of the non-athletic population may require the re-evaluation of their vitamin B_12_ status.

In our opinion and considering the aforementioned doubts concerning the reliability of blood vitamin B_12_ tests alone, such a re-evaluation is advisable [17,19,56,57,58,59]. Bearing in mind potential hematologic benefits, athletes with a vitamin B_12_ concentration of 200–400 pg/mL require special attention, wider monitoring and personalized supplementation. An improvement of the RBC parameters was found at the concentrations above 395 and 488 pg/mL, depending on the statistical method. Therefore, in the authors’ opinion, the lower limit of normal range for vitamin B_12_ in athletes should be set at approx. 400 pg/mL and everyone below that level should be additionally monitored, consulted by a dietician and supplemented.

The available data on treatment with vitamin B_12_ show that oral cyanocobalamin at the doses of 1000–2000 μg daily is sufficient and as effective as intramuscular injections, regardless of the etiology of the deficiency [70,71,72]. Alternatively, the nasal route of cobalamin administration can also be considered [73,74]. Although the aforementioned data refer to the sedentary population (mainly older people), regularly monitored and orally supplemented athletes should not have any problems achieving the target zone of approx. 400 pg/mL. Thus, the wide use of injections is not justified and it is rather unfortunate that, despite the apparently weak evidence for its ergogenic effects, 34% of athletes had declared the use of vitamin B_12_ injections according to the present study. Interestingly, 13% who did not believe in effectiveness of B_12_ still used the injections. The latter method is more popular among the endurance athletes (46% vs. 30% in strength group), which supports the hypothesis about a strong belief in possible enhancement of red blood cell formation by vitamin B_12_.

Intramuscular injections resulted in the vitamin B_12_ concentration of over 1000 pg/mL already within a week, and after four weeks the value reached almost 3000 pg/mL [75]. Therefore, injections in athletes should only be applied in ‘acute’ cases without sufficient time (a month) to be treated orally. A regular intramuscular treatment is not justified as it poses the unnecessary risk of achieving an extremely high vitamin B_12_ concentration.

In the present study, the tendency of improvement in the RBC parameters persisted up to 700 pg/mL and no additional benefit was observed at higher levels. A vitamin B_12_ concentration above 700 pg/mL was found in 38% of samples and those athletes should be advised to revise the indications to vitamin B_12_ supplementation and consider it again only when the blood concentration approaches 400 pg/mL.

## 5. Conclusions

Athletes should regularly monitor their blood vitamin B_12_ concentration and, if necessary, adjust the oral supplementation individually to achieve the zone of 400–700 pg/mL. Special attention is required in vegetarians, vegans and athletes with a low (but within the normal range) vitamin B_12_ concentration (200–400 pg/mL). Bearing in mind the potential benefits of the improvement in the red blood cell parameters, athletes with insufficient vitamin B_12_ concentration should be supplemented.

## Figures and Tables

**Figure 1 nutrients-12-01038-f001:**
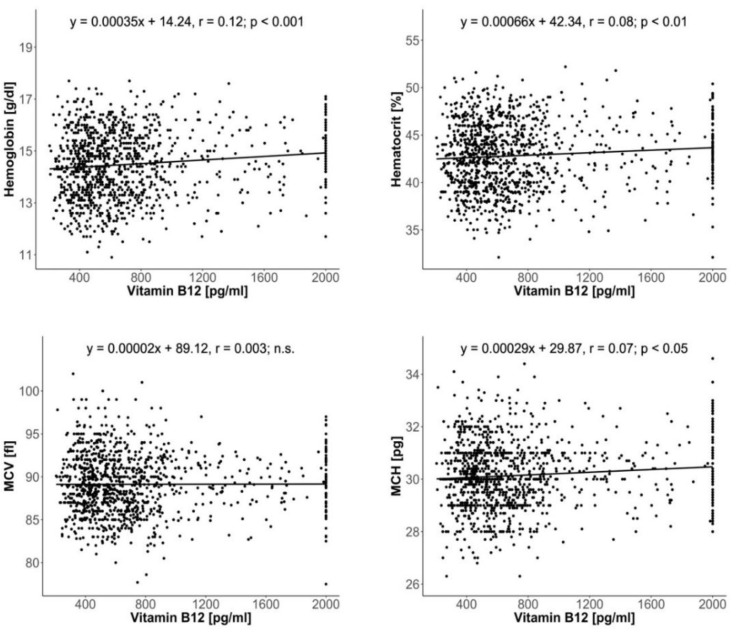
The correlations between total serum vitamin B_12_ concentration and hematological indices: hemoglobin concentration, hematocrit, mean corpuscle volume (MCV) and mean corpuscle hemoglobin (MCH).

**Figure 2 nutrients-12-01038-f002:**
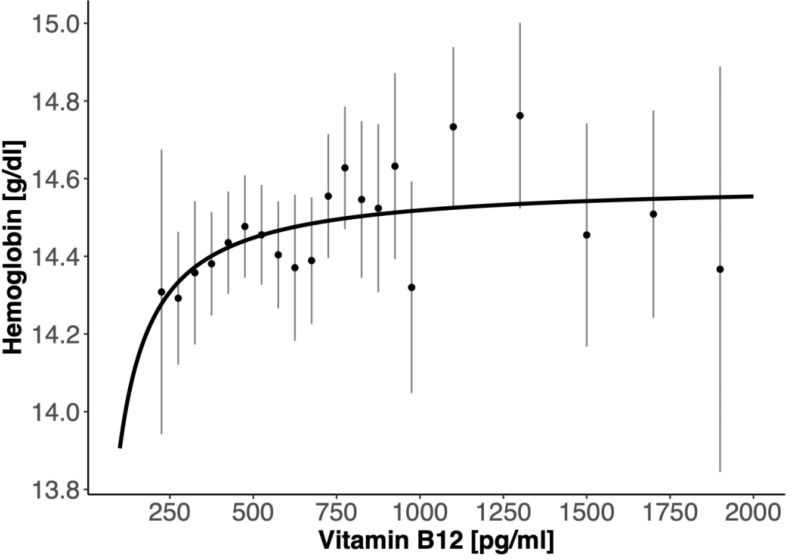
The fitted saturation curve for hemoglobin used to calculate the threshold value for the 99% saturation.

**Figure 3 nutrients-12-01038-f003:**
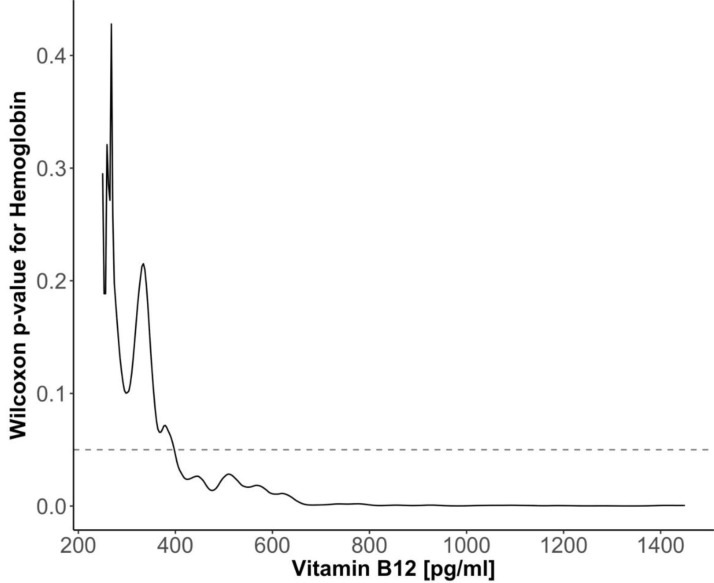
The Wilcoxon *p*-value for the significance of differences in hemoglobin concentration in samples continuously divided according to different vitamin B_12_ concentration (floating B_12_ cut-off), dashed line designates the level of significance 0.05.

**Figure 4 nutrients-12-01038-f004:**
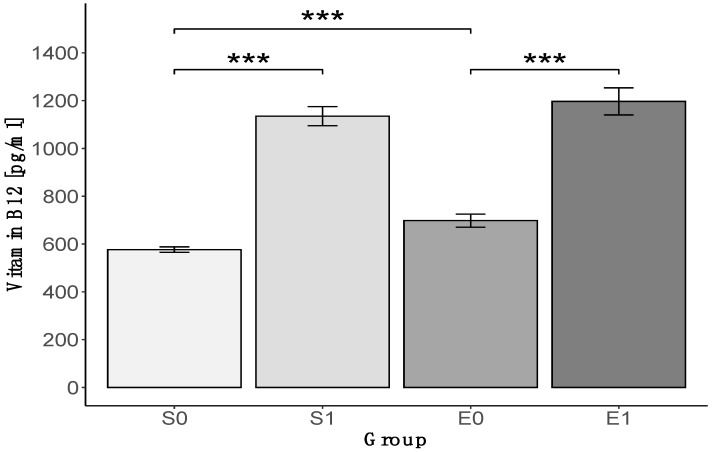
The vitamin B_12_ concentration in blood in strength (S) and endurance (E) athletes, who either declared the use of vitamin B_12_ injections (S1 and E1) or no use of injections (S0 and E0); *** *p* < 0.001.

**Figure 5 nutrients-12-01038-f005:**
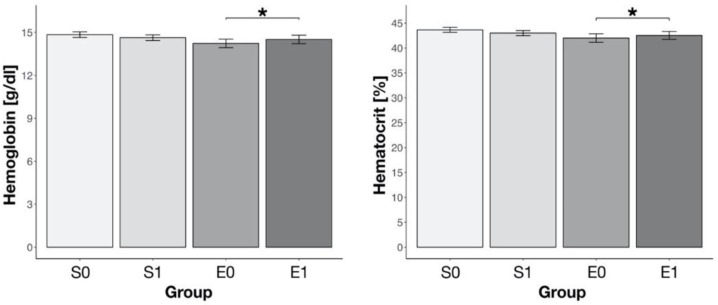
The hemoglobin concentration and hematocrit in strength (S) and endurance (E) athletes according to the use of injections (0–without injection; 1–after injection), * *p* < 0.05.

**Table 1 nutrients-12-01038-t001:** The number and percentage of samples of the strength and endurance athletes with vitamin B_12_ concentration below 300, 350 and 400 pg/mL and above 700 pg/mL.

Vitamin B_12_	<300 pg/mL	<350 pg/mL	<400 pg/mL	>700 pg/mL
Strength *N* (%)	48 (5.3%)	103 (11.4%)	186 (20.6%)	296 (32.7%)
Endurance *N* (%)	2 (0.9%) **	7 (3.1%) ***	19 (8.4%) ***	128 (56.4%) ***
Total *N* (%)	50 (4.4%)	110 (9.7%)	205 (18.1%)	424 (7.5%)

* the difference between strength and endurance groups in vitamin B_12_ concentration, ** *p* < 0.01, *** *p* < 0.001.

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
