# Peer review of "Vitamin B12 Status and Optimal Range for Hemoglobin Formation in Elite Athletes"

_nutrients, 2020, doi:10.3390/nu12041038_

Round 1
Reviewer 1 Report
General comments.
The revised and resubmitted manuscript by Krzywański et al “Vitamin B12 status and optimal range for hemoglobin formation in elite athletes” shows a considerable improvement in comparison to the original version. A few issues should be, nevertheless, corrected (including language). A number of suggestions are listed in Specific and Minor comments, as well as directly in the edited pdf-file. The manuscript can be accepted after correction of these shortcomings.
Specific comments.
1. The text right above Figure 2. The authors perform numerical estimation of inflection points for the function in Figure 2. Yet, I have reasonable doubts about the accuracy of such approach. Thus, the hyperbolic function used for fitting has neither maximums / minimums nor inflection points, because its first, second etc derivatives do not equal zero at any point. I am afraid that the authors have used an incorrect setup when running a numerical estimator program. Any mentioning of inflection points of the Function 1 should be deleted from the text.
2. The text right above Figure 3. I have earlier expressed my concerns about validity of analysis in Figure 3. As the author prefer to present it anyway, I suggest to add a comment “It should be, however, noted that the interpretation of floating cut-off analysis should be considered with some caution. Indeed, such analysis is affected by clustering of the points on the surface, which changes from dataset to dataset and is irrelevant for the overall dependence of Hb on B12.”
Minor comments.
1. The symbol plus-minus “±” is consistently absent in the text. The authors should use a standard font type for this symbol.
2. The authors use the term “vegetarians”. Please, clarify whether you mean vegetarians or vegans or both?
3. Multiple language issues are corrected directly within the document, see the attached file.
Author Response
We wish to thank for the effort reviewing our manuscript. The paragraph with demographic data which is at the beginning of the methods was unintentionally repeated below Table 1. We failed to notice that but it has been deleted now.
General comments
We are most thankful for the valuable comments and the impressive number of language corrections, which of course were applied to the manuscript.
Specific comments
Re. 1. Thank you for your remark. You are right that the function has no analytical inflection point. We used numerical estimator, which, in this setup, established the point geometrically based on polynomial estimations. However, as the method is designed to find true inflection point in noisy data, the use of such value to state the conclusions may be questionable. Therefore, we decided to use your suggestion and delete this approach from the text.
Re. 2. Following the Reviewer’s remark, the suggested comment has been introduced into the manuscript.
Minor comments
Re 1. The ± was used in the standard manner, we are astonished how it was deformed and obviously corrected it.
Re. 2. We meant both vegans and vegetarians, which has been clarified in the text.
Re. 3. We are again thankful for the impressive number of language corrections, which have of course been applied in the manuscript. We have left the “track and field” in the methods, as it defines exactly the group of subjects as athletics; leaving only the term “athletes” would have a wider meaning i.e. sportsmen of all sports, which is what we wanted to avoid. The use of SEM is described in the methods, that is why we do not repeat that information in the results and also unified the abbreviation (SEM and not the SE).
Reviewer 2 Report
The work presented by Jarosław Krzywański and colleagues assessed the vitamin B12 status in Polish elite athletes and its influence on red blood cell parameters. This study is novel, and the manuscript is well written and very well organized. However, I just have some minor concerns.
- There are few typo mistakes like- in abstract line 6: “An average vitamin B12 concentration in all subjects was….”, page 2/line 4: “when the total serum vitamin B12….”.
- Introduction section, please write more about how B12 is involved in methylation reaction of DNA, RNA, and protein.
- Introduction section: Page 2/4th paragraph: if it is journal's style to write purpose as a separate heading under the introduction section then it is fine, otherwise you do not need to write it as a different heading.
- In the methods section, it would be easier, if you could put a table on how you classified your study subjects.
Author Response
We wish to thank for the effort reviewing our manuscript. The paragraph with demographic data which is at the beginning of the methods was unintentionally repeated below Table 1. We failed to notice that but it has been deleted now.
Re. 1. To our surprise, the typos sneaked in while opening the document on the other computer. Naturally, they have been corrected throughout the whole manuscript (especially ±).
Re. 2. The sentence concerning the involvement of B12 in methylation reactions was expanded according to the request.
Re. 3. The separate heading “Purpose” was created according to the older sample article we have found in the journal. We have checked the recent articles and the heading is not used there. Therefore, following Your suggestion, it has been deleted. Thank you for remark.
Re. 4. As the first approach we did it in the table, just as Reviewer suggested (all subjects above and then groups S0, S1, E0 and E1 below), but due to a long text boxes and many columns, it didn’t turn out as clear as we had expected. That is why we decided to put it in the text and would prefer to leave it this way.